# Transfer dynamics of multi-resistance plasmids in *Escherichia coli* isolated from meat

**Tania S. Darphorn[1], Belinda B. Koenders-van Sintanneland[1], Anita E. Grootemaat[2], Nicole N. van der Wel[2], Stanley Brul[1], Benno H. ter Kuile** [1,3]*

**1** Laboratory for Molecular Biology and Microbial Food Safety, Swammerdam Institute for Life Sciences, University of Amsterdam, Amsterdam, The Netherlands, **2** Department of Cell Biology and Histology, Electron Microscopy Centre Amsterdam, Academic Medical Center, Amsterdam, The Netherlands, **3** Netherlands Food and Consumer Product Safety Authority, Office for Risk Assessment, Utrecht, The Netherlands

* b.h.terkuile@uva.nl

**Data Availability Statement:** All relevant data are within the paper.

**Funding:** This study was financed by a grant from the Netherlands Ministry of Agriculture to BHTK.

## Abstract

Resistance plasmids are crucial for the transfer of antimicrobial resistance and thus form a matter of concern for veterinary and human healthcare. To study plasmid transfer, food-borne *Escherichia coli* isolates harboring one to five known plasmids were co-incubated with a general recipient strain. Plasmid transfer rates under standardized conditions varied by a factor of almost $10^6$, depending on the recipient/donor strain combination. After 1 hour transconjugants never accounted for more than 3% of the total number of cells. Transconjugants were formed from 14 donors within 1 hour of co-incubation, but in the case of 3 donors 24 hours were needed. Transfer rates were also measured during longer co-incubation, between different species and during repeated back and forth transfer. Longer co-incubation resulted in the transfer of more types of resistance. Maximum growth rates of donor strains varied by a factor of 3. Donor strains often had higher growth rates than the corresponding transconjugants, which grew at the same rate as or slightly faster than the recipient. Hence, possessing one or more plasmids does not seem to burden the harboring strain metabolically. Transfer was species specific and repeated transfer of one plasmid did not result in different transfer rates over time. Transmission Electron microcopy was used to analyze the morphology of the connection between co-incubated strains. Connection by more pili between the cells resulted in better aggregate formation and corresponded with higher transfer rates.

## Introduction

Resistance plasmids play a crucial rule in the spread of antimicrobial resistance in veterinary and human healthcare. *Escherichia coli* infections are becoming increasingly untreatable due to the encoded resistance genes on plasmids such as extended spectrum beta-lactamases (ESBL), beta-lactamases (BL) and tetracycline resistance, especially observed in livestock [1–4]. Pathogens resistant to antimicrobials can transfer from animals directly to farmers [5, 6] or from foodstuffs to human consumers [7]. All applications of antibiotics in farming and human

**Competing interests:** The authors have declared that no competing interests exist.

healthcare can increase the spread of resistance plasmids and genes [8]. Plasmids are able not only to spread between cells of the same species but also to different species of bacteria [9–14]. To better understand the spreading of resistance mediated by plasmids, we explored the dynamics of conjugation and plasmid transfer.

The conjugation machinery encoded on plasmids enables them to transfer from cell to cell [15–17], but conjugation has been shown to be mostly species specific [10, 12]. Conjugation rates often remain low as the process has a high metabolic cost [18]. There are multiple conjugative systems, which have been used to characterize plasmid types in combination with their incompatibility groups [14]. The incompatibility groups IncI, IncF and IncX all have their own type of conjugative system that transfers plasmids at different rates [9, 14, 15, 19].

Resistant *E. coli* strains isolated from meat destined for the consumer market often contain multiple plasmids in a single strain [20]. These plasmids consisted of several types, but the IncI, IncX and IncF types were most present in these isolates. The plasmids possessed a wide variety of resistance genes between them. This raises several questions: what effect does the cellular presence of multiple plasmids have on conjugation? Is transfer with cells harboring multiple plasmids affected by length of the co-incubation or co-incubation with different species? A final question is whether continuous back and forth transfer of one such naturally occurring plasmid influences the efficiency of the transfer process.

To answer these questions, a standard transfer method was designed to measure actual transfer events while excluding the effects of competition and growth. A set of donor *E. coli* isolated from foodstuffs, is co-incubated with a general recipient to obtain transconjugants. Transfer rates are correlated to the growth rates of the donor, recipient and transconjugant strains. The transfer method was adapted to also include longer co-incubation, different species and continuous transfer as variables. Lastly, transmission electron microscopy was used to examine the morphology of pili between the co-incubated strains.

## Materials & methods

### Bacterial strains, solutions, MICs and growth rate

All strains used are shown in Table 1. The *Escherichia coli* donor strains were isolated from foodstuffs by the Dutch Food and Consumer Product Safety Authority (NVWA) and characterized by Wageningen Bioveterinary research (WBVR). Dr. Kees Veldman of WBVR selected and donated 28 strains that were known to contain plasmids and that originated from turkey, bovine or chicken meat. Out of this set 17 selected for having at least beta-lactam resistance were used in this study. The NVWA originally established the resistance with a sensitivity test [21] and this was confirmed afterwards using MIC assays [22]. Plasmid presence was initially confirmed by the detection of incompatibility groups [23] and later confirmed by plasmid isolation and long and short read sequencing [20]. These sequences were analyzed using CGEs PlasmidFinder 2.1 [24] and ResFinder 4.0 [25] to determine and confirm and find other incompatibility groups and resistance genes. The chloramphenicol resistant (chlor$^R$) *E. coli* MG1655 YFP (kindly provided by MB Elowitz [26]) was used as common recipient strain in transfer experiments. Another *E. coli* MG1655 strain was evolved to obtain enrofloxacin resistance by stepwise increasing the antibiotic concentration. This strain was used as a second standard recipient whenever the donor strain already expressed chloramphenicol resistance. Recipients were chosen so that the donor could not grow on the plate selecting for the transconjugant. Another *E. coli* strain was used for continuous back and forth transfer experiments: *E. coli* JW3686. This strain is an indole knockout mutant with inserted kanamycin resistance (kan$^R$). Three other organisms were used as recipient strains in experiments to examine

**Table 1. List of *E. coli* strains used as donors, names of recipient strains and their chromosomally encoded antibiotic resistance and characterization of plasmids as found within the donor strains, showing number of plasmids (#), replicon types and beta-lactamase genes.** (Sequences can be found at Genbank accession numbers MW390511 to MW390552).

| Donor | Recipient | | | Donor plasmids | |
|---|---|---|---|---|---|
| Strain | Strain | resistance | # | Replicon type | Beta-lactamase |
| *E. coli* 2073 | *E. coli* MG1655 | chlor$^R$ | 2 | IncI1, IncX4 | $bla_{CTX-M-1}$ |
| *E. coli* 2082 | *E. coli* MG1655 | chlor$^R$ | 5 | IncI1, IncX4, IncX1, IncFIB/FII, p$_{phage}$ | $bla_{SHV-12}$, $bla_{TEM}$ (3x) |
| *E. coli* 3153 | *E. coli* MG1655 | chlor$^R$ | 3 | IncI1, IncX4, IncFII | $bla_{CTX-M-1}$ |
| *E. coli* 3156 | *E. coli* MG1655 | chlor$^R$ | 3 | IncB/0/K/Z, IncX4, IncFIB/FII | $bla_{CMY-2}$, $bla_{TEM-1B}$ |
| *E. coli* 3170 | *E. coli* MG1655 | chlor$^R$ | 3 | IncI1, IncFIB, IncFIC/FII | $bla_{CTX-M-1}$ |
| | *P. aeruginosa* POA-1 | | | | |
| | *P. aeruginosa* PA-1 | | | | |
| | *E. faecalis* OG1RF | | | | |
| *E. coli* 3171 | *E. coli* MG1655 | chlor$^R$ | 2 | IncI1, IncFII | $bla_{CMY-2}$ |
| *E. coli* 3182 | *E. coli* MG1655 | chlor$^R$ | 3 | IncN, IncFII, IncFIB | $bla_{CTX-M-55}$, $bla_{TEM-1B}$ |
| *E. coli* 3203 | *E. coli* MG1655 | chlor$^R$ | 2 | IncI, IncFII | $bla_{CTX-M-8}$ |
| *E. coli* 3215 | *E. coli* MG1655 | chlor$^R$ | 4 | IncI2, IncX4, IncFIB/FIC, p$_{phage}$ | $bla_{TEM-52c}$ |
| *E. coli* 3227 | *E. coli* MG1655 | chlor$^R$ | 2 | IncI1, IncFIB/FII | $bla_{CTX-M-2}$, $bla_{TEM-1B}$ |
| *E. coli* 3231 | *E. coli* MG1655 | chlor$^R$ | 3 | IncN, IncFIB/FII, p0111 | $bla_{TEM}$, $bla_{SHV-12}$ |
| *E. coli* 3277 | *E. coli* MG1655 | enr$^R$ | 1 | IncFIB/FIC | $bla_{CTX-M-55}$ |
| *E. coli* 3284 | *E. coli* MG1655 | enr$^R$ | 1 | IncFIB/FII | $bla_{CMY-2}$* |
| *E. coli* 3301 | *E. coli* MG1655 | chlor$^R$ | 3 | IncI1, IncX1, IncFIB/FII | $bla_{CTX-M-1}$ |
| *E. coli* 3308 | *E. coli* MG1655 | enr$^R$ | 4 | IncI1, IncY, IncFIB, IncFIC/FII | $bla_{CTX-M-1}$ |
| *E. coli* 3310 | *E. coli* MG1655 | chlor$^R$ | 2 | IncB/0/K/Z, IncFIB/FIV | $bla_{CMY-2}$ |
| *E. coli* 3334 | *E. coli* MG1655 | chlor$^R$ | 2 | IncX1, col156 | $bla_{CTX-M-32}$ |
| *E. coli* MG1655 pIncI3170 | *E. coli* JW3686 | kan$^R$ | 1 | IncI1 | $bla_{CTX-M-1}$ |
| *E. coli* JW3686 pIncI3170 | *E. coli* MG1655 | chlor$^R$ | 1 | IncI1 | $bla_{CTX-M-1}$ |

*Beta-lactamase is unconfirmed if located in the plasmid or the chromosome

species specificity of the transfer process: *Pseudomonas aeruginosa* POA-1 and PA-1, as well as *Enterococcus faecalis* OG1RF.

Defined minimal mineral medium containing 55 mM glucose with a pH of 6.9 and a buffer of 15.6 g/L Na2H2PO4 [27] was used to grow strains overnight at 37°C degrees and shaken at 200 rpm. The same minimal medium without glucose was used for standard transfer experiments to prevent confounding of the outcome by growth. Selective plates were made with Luria broth (LB) (1% NaCl; 0.5% yeast extract; 1% bactotryptone) with 2% agar and appropriate antibiotic stocks. Stock solutions of 10 mg/ml antibiotics for ampicillin, chloramphenicol, enrofloxacin, kanamycin and tetracycline were filter sterilized and stored at 4°C for up to 2 weeks maximum. The final concentration of antibiotics in the selective plates were set to 64 µg/ml.

MIC and growth rates of the donor, recipient and transconjugant strains were measured as described by Schuurmans, Nuri Hayali [22] in 96-well plates in a ThermoScientific Multiskan FC spectrophotometer plate reader. Plates were shaken and kept at 37°C in a final volume of 150 µl with a starting OD595 of 0.05 for 23 hours. Antibiotic concentrations increasing by a factor of 2 and ranging from 1 µg/ml to 2048 µg/ml were used. The lowest concentrations that limited final OD to 0.2 or less was reported as MIC. The maximum specific growth rate (µmax) was determined out of the growth curves that were obtained from 8 wells with minimal medium. This method was used for growth rate determination, because of the high reproducibility.

## Transfer experiments

For most transfer experiments a standard method was established to reduce the influence of other parameters, such as growth and increase reproducibility. Cells of an overnight culture were spun down at 4400 rpm for 15 minutes, the old medium was removed and replaced with minimal medium without glucose. After careful mixing, the cells were starved for 4 hours at 37˚C and shaken at 200 rpm. To start the transfer experiment, the starved donor and recipient strains were mixed at a 1:1 ratio with a final OD600 of 0.25 in minimal medium without glucose. Preliminary experiments showed that after 1 and 24 hours of co-incubation the total cell number remained stable and that the cells were still viable. The same preliminary experiments also showed that the transfer was either very rapid, with many transconjugants after 1 hour, or very slow, with only a few after 24 hours. Therefore these two time points were used throughout the study. Total numbers of recipient, donor and transconjugant cells were determined by plating dilutions on appropriate selective LB agar plates. Plates to select for transconjugants contained both antibiotics that the donor and recipient strain respectively were resistant to. Strains were identified as transconjugants with MIC assays and the presence of the plasmid in the transconjugants was confirmed by isolating the plasmid and sequencing it entirely (Genbank accession numbers MW390511 to MW390552). 2 or 3 transconjugants were isolated from the selective plates and tested for MIC and growth rate and stored in glycerol stocks at -80˚C. In some experiments, this standard method was adapted according to the experimental design. All transfer experiments were performed in triplicate.

*E. coli* 3170 was attempted to mate with other species (*P. aeruginosa* and *E. faecalis*) in multiple transfer experiments. Apart from the standard method, a second method was used with minimal medium with added glucose and a third method with the use of LB medium instead of minimal medium for 1 and 24 hours. Other than the difference in medium the set-up of the experiment stayed the same. In an additional experiment, a 1:1 mixed solution of recipient and donor strains with an OD600 of 1 in an Eppendorf tube was directly plated on selective plates.

One plasmid was transferred multiple times back and forth between 2 recipient strains. A transconjugant containing the IncI1 plasmid of *E. coli* 3170 (*E. coli* MG1655 pIncI3170) was mated with the standard recipient using the standard method and with *E. coli* JW3686 (kan[R]) with glucose in the medium as this recipient otherwise did not survive the experiment. To continue the transfer of the plasmid, a transconjugant was isolated and grown overnight from the 1-hour co-incubation plates and used as the new donor strain (*E. coli* JW3686 pIncI3170) in a new mating with *E. coli* MG1655 (chlor[R]) as recipient. In the single case that the 1-hour plates did not yield any transconjugants, a transconjugant from the 24-hour plates was isolated instead. These steps were repeated 11 to 12 times. Transconjugant strains were tested for MIC and indole to ascertain that they were true transconjugants. The indole test discriminated the indole knockout strain *E. coli* JW3686 from *E. coli* MG1655, as the former turns yellow when mixed with Kovac's reagent while the latter will color pink for the presence of indole.

## Transformation

Transformations were attempted by mixing competent *E. coli* MG1655 with either plasmid DNA or dead cells from *E. coli* 3153 and 2073. Plasmid DNA was isolated using the Qiagen Plasmid Maxi Kit. Dead cells were obtained by pasteurizing cells at 72˚C for 15 seconds. The transformation was done by heat shock at 42˚C for 45 seconds followed by cold shock on ice for a maximum of 5 minutes for plasmid DNA and 30 minutes for dead cells. Before plating the transformed cells on selective plates, the cells were recovered in S.O.C. medium (2% Tryptone; 0,5% yeast extract; 0,4% glucose; 10 mM NaCl; 2,5 mM KCl; 10 mM MgCl$_2$) for 60 minutes for plasmid DNA and 2 or 5 hours for dead cells at 37˚C and 200 rpm. The dead cells

were additionally transformed without the heat and cold shocks. Transformations with water and a control plasmid (pUC19) were performed as a negative and positive control, respectively.

## Transmission electron microscopy

Pilli of *E. coli* 3153, 2073, MG1655 and JW3686 were visualized using transmission electron microscopy (TEM). Part of the fresh overnight cultured cells were fixed with 4% paraformaldehyde (PFA) and allowed to slowly settle for 4 hours and transported and stored at 4°C. Just before EM visualization fixed and unfixed cells were contrasted on a copper formvar coated grid using uranyl acetate. Fresh non-fixed cells showed a better result than PFA fixed cells as these did not show as much or any pili at all. Fresh cells were therefore used for the final analysis. Both single and mixed populations were visualized using TEM. The mixed populations consist of a 1:1 ratio of a donor and recipient strain and were co-incubated for no longer than 2 hours before storing and transportation at 4°C. Visualization and handling of the TEM (Tecnai T12 at 120 kV with a Veleta digital camera) was performed at the Electron Microscopy Centre Amsterdam (EMCA, Amsterdam UMC).

## Results

Transfer experiments were conducted to determine the rates at which transconjugants can be formed when two potential mates were co-incubated. The mating procedure was designed to eliminate the effect of growth and competition, so that the numbers correspond to actual transfer events. Transfer rates ranged between 0,2 to approximately 30.000 transconjugants per million cells per hour ([Fig 1]). These numbers corresponded at maximum to 3% of the total amount of cells. Three donor strains were never able to produce transconjugants after 1 hour but did form them after 24 hours. One strain did not form any transconjugant at all (not

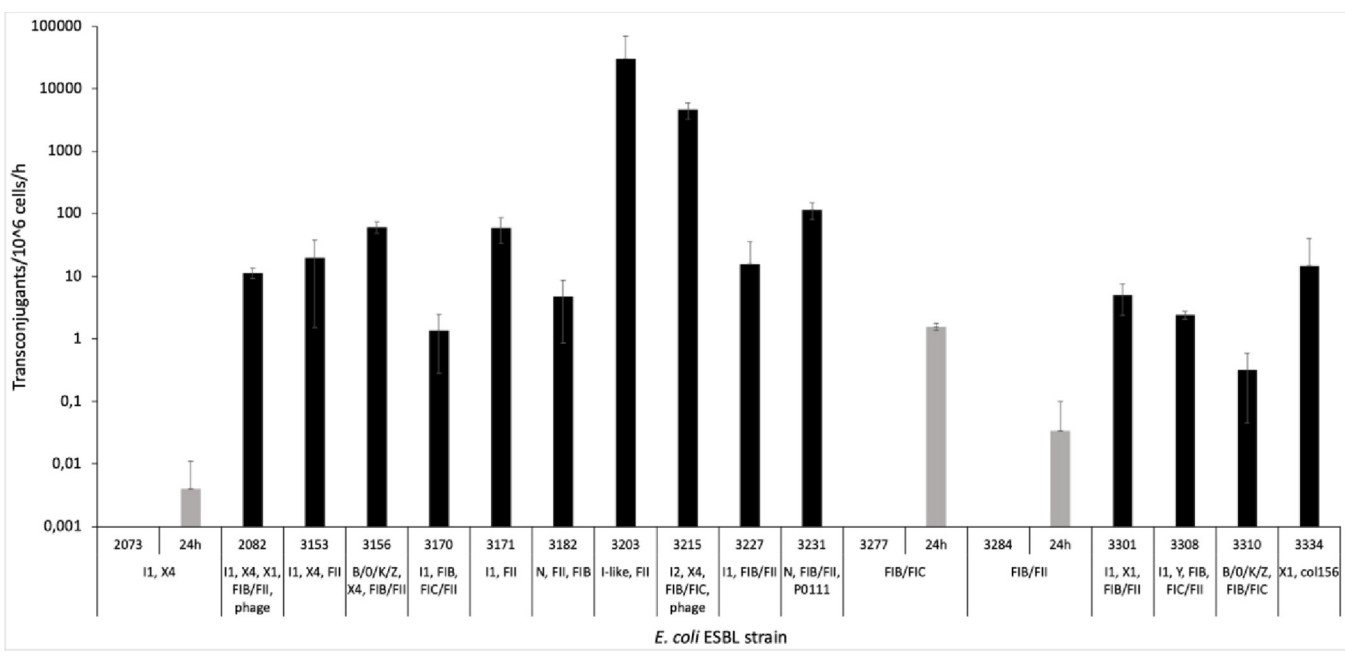

**Fig 1. Transconjugants/10^6 cells/hour per strain in log scale with standard deviation and the corresponding plasmid types as present in the donor strain.** All measurements are as shown after 1 hour. When no transconjugants were obtained after 1 hour, the 24-hour sample is also shown.

shown in the graphs). Two of the three strains that transferred resistance only after 24 hours contained only a single plasmid, which belonged to the IncF type. Strain 2073 was the exception which contained both a IncI and IncX4-type plasmids, but still transferred at a low rate. All other strains that contained an IncI-like plasmid transferred plasmids within 1 hour, but the transfer rates varied highly with no discernable trend. As control experiments, some transfers were performed with different selective plates, using tetracycline instead of ampicillin, to see if tetracycline harboring plasmids could also be selected for. Standard methods and methods with LB and added glucose did not yield any transconjugants for the tested donor strains during 1 hour or 24 hour incubations.

Transformation was performed as a control experiment with two strains to determine whether transformation of large plasmids was possible, thus assessing a possible role of transformation as a factor influencing transfer rates. Transformation for both strains 3153 (fast transfer) and 2073 (slow transfer) showed no plasmid uptake using both dead cells or pDNA in different media (LB and SOC). A control plasmid (pUC19) was able to be transformed into the competent cells.

Possessing plasmids is supposed to have a metabolic burden, lowering the growth rate. The maximum specific growth rates of the donor strains ranged from 0,4 to 1,7 h$^{-1}$ (Fig 2). The growth rates of the transconjugants varied far less and equaled minimally that of the recipient cells (0,42 h$^{-1}$), but mostly exceeded it by a factor of maximally 1.6. This observation suggests that the metabolic burden was not pertinent in these transconjugants.

Transfer between widely different species was examined using *Pseudomonas aeruginosa* and *Enterococcus faecalis* as recipients and *E. coli* ESBL 3170 as donor. Strain ESBL 3170 was selected because it contained plasmid types that were found in a wide variety of bacterial species. Multiple attempts under different conditions did not yield any transconjugants (Table 2). The methods of transfer attempted were the standard method, addition of glucose, using LB medium instead of Evans minimal medium and direct plating in high concentrations as well as using longer incubation times. The donor strain had an average transfer rate when co-incubated with the standard *E. coli* recipient.

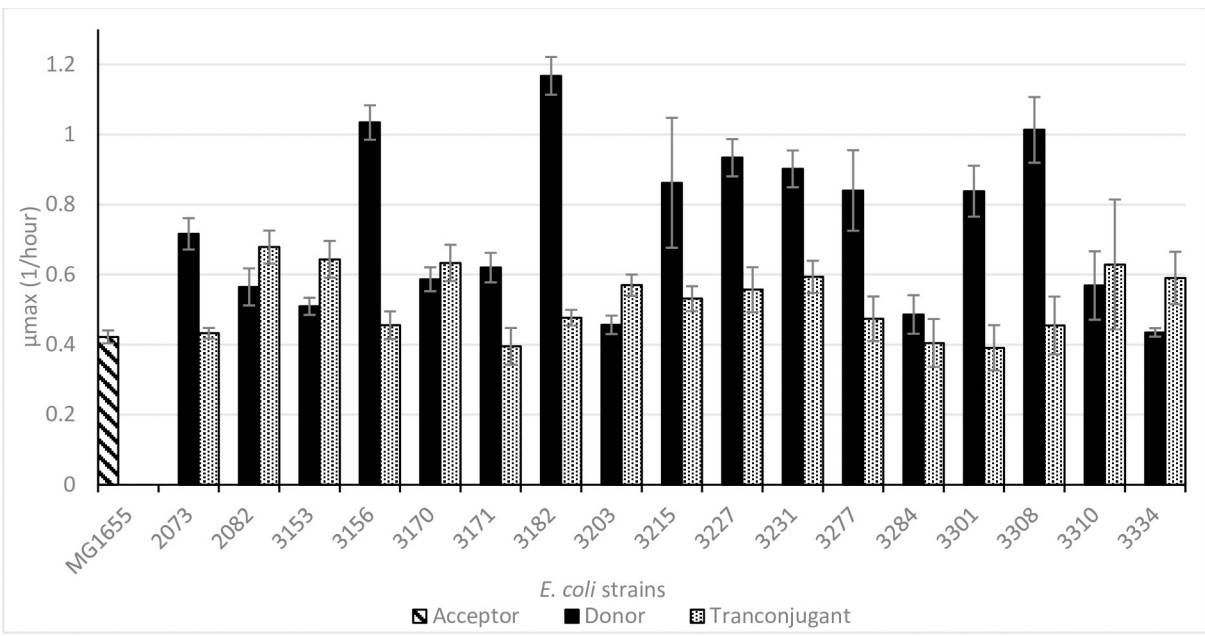

**Fig 2. Average $\mu_{max}$ (h$^{-1}$) per recipient (diagonal stripes), donor (solid) and transconjugant (dots) strain with standard deviation.** Average values were determined from at least 2 replicates, that individually were measured 8 times.

**Table 2. Number of transconjugants found for transfer with *E. coli* ESBL 3170 as donor and *E. faecalis* OG1RF and two *P. aeruginosa* strains as recipient.**

| Species | Transconjugants/million cells |
|---|---|
| *E. faecalis* OG1RF | 0 |
| *P. aeruginosa* PAO-1 | 0 |
| *P. aeruginosa* PA-1 | 0 |

(This table can be omitted without losing information, but will be useful for a reader who only glances at the article instead of reading it thoroughly.)

To examine whether the transfer rate would increase after repeated transconjugation because the plasmid could adapt to it, a transfer experiment was conducted consisting of repeated back and forth transfer of one specific plasmid. Contrary to our expectation this did not influence transfer rate nor resistance. These transfers were performed using the standard method (Fig 3A) and were repeated with the addition of glucose (Fig 3B) for 12 or 11 consecutive transfers, respectively. The transconjugants' MICs for ampicillin, kanamycin and chloramphenicol did not change throughout the transfers. This outcome suggests that the combination of donor and recipient determined the transfer efficiency.

Many donor strains harbor more than one plasmid, see Table 1. If a donor strain harbors several plasmids, multiple or different plasmids could be transferred at a time. To see whether there was any difference in transferred resistance all transconjugants from transfer with donor strain 2082 that harbored 5 plasmids were isolated and tested for the MIC for ampicillin, chloramphenicol (Fig 4A), tetracycline (Fig 4B) and kanamycin (Fig 4C). The resistance for both chloramphenicol and ampicillin stayed similar over 24 hours as expected because of the selective plates. Higher resistance levels are reached for both tetracycline and kanamycin after 24 hours co-incubation compared to after 1 hour. This suggests that more plasmids were transferred during prolonged co-incubation.

## Electron microscopy

TEM images were made from cells to determine whether any structural differences could be observed between a slow donor, a fast donor and the recipient strain. Two donor strains were compared: *E. coli* 2073, because it transferred only after 24 hours, and *E. coli* 3153 which transferred faster. Similarities were detected in shape and size of the cells, as well as the presence of a single flagellum. Differences between the two strains were detected in the number of pili and the formation of cell-aggregates. *E. coli* 2073 showed some small pili (Fig 5A) whereas *E. coli* 3153 showed more pili (Fig 5B). *E. coli* MG1655 carried several flagella per cell (Fig 5C), where *E. coli* JW3686 contained only 1 or no flagellum (Fig 5D). Aggregates of *E. coli* 3153 showed more cells, both alone and in co-incubation with *E. coli* MG1655 (Fig 5E) and cells were covered by many pili. The pili, but also some flagella, seem to connect to a neighbouring cells' membrane, possibly enabling transfer of plasmids. As these cells are dried and thus flattened out on the formvar film coating on the EM grid, it is impossible to confirm actual fusion between pili and the recipient cell. However, as multiple pili were bridging the space between two cells, it suggests a possible mechanism for transfer of plasmids.

## Discussion

In order to ensure that the experiments are comparable to each other the effect of growth during the incubations was eliminated by the absence of a carbon and energy source. As a result the absolute numbers of plasmid transfers in this study were comparatively low. Newly formed

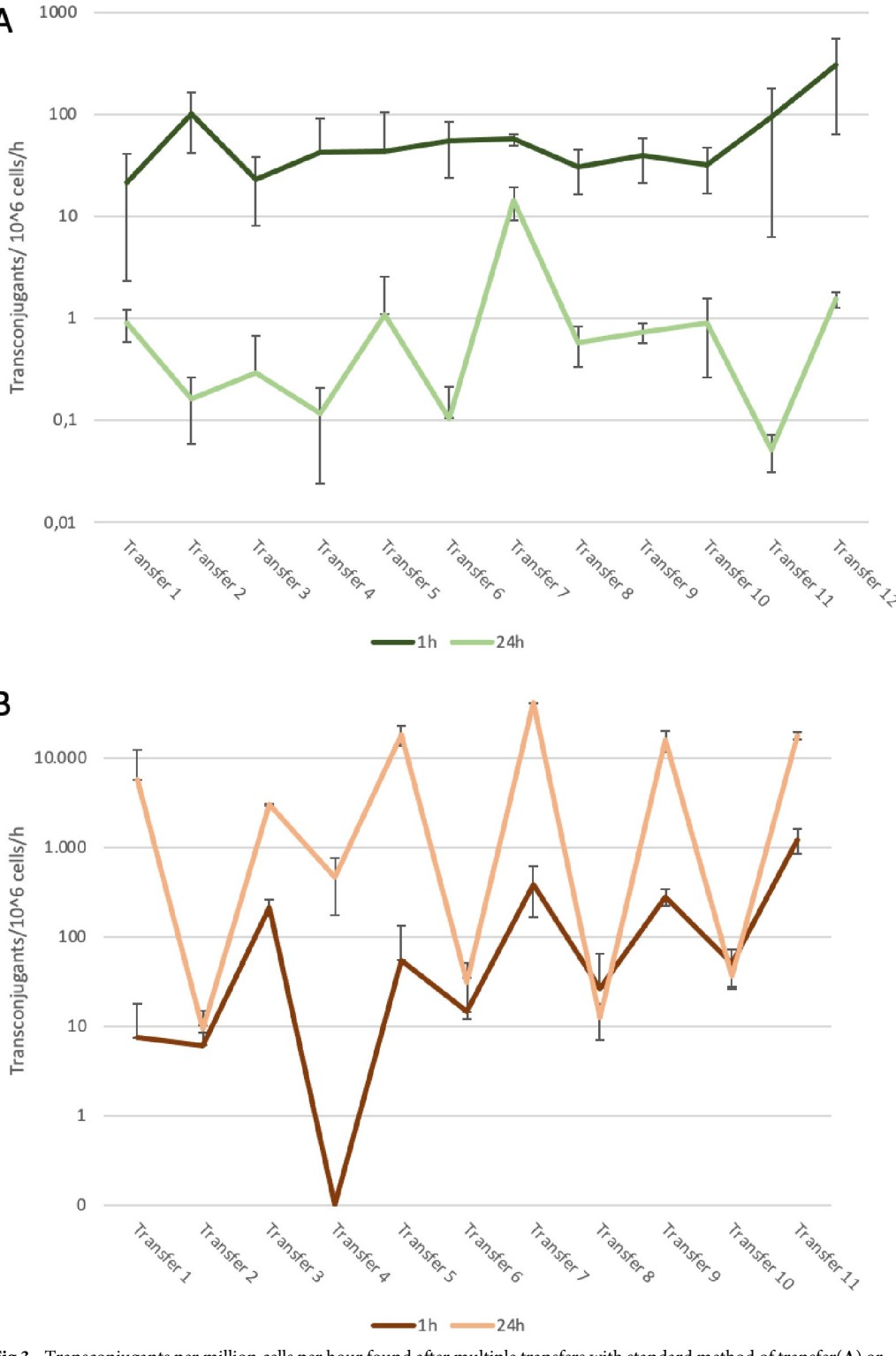

**Fig 3.** Transconjugants per million cells per hour found after multiple transfers with standard method of transfer(**A**) or addition of glucose (**B**) measured after 1 and 24 hours. Two recipient s were used: *E. coli* JW3686 with kanamycin resistance (uneven transfers) and *E. coli* MG1655 with chloramphenicol resistance (even transfers). The obtained transconjugants were used as a donor strain for the next transfer round. Standard deviations are shown but are distorted because of the use of the log scale.

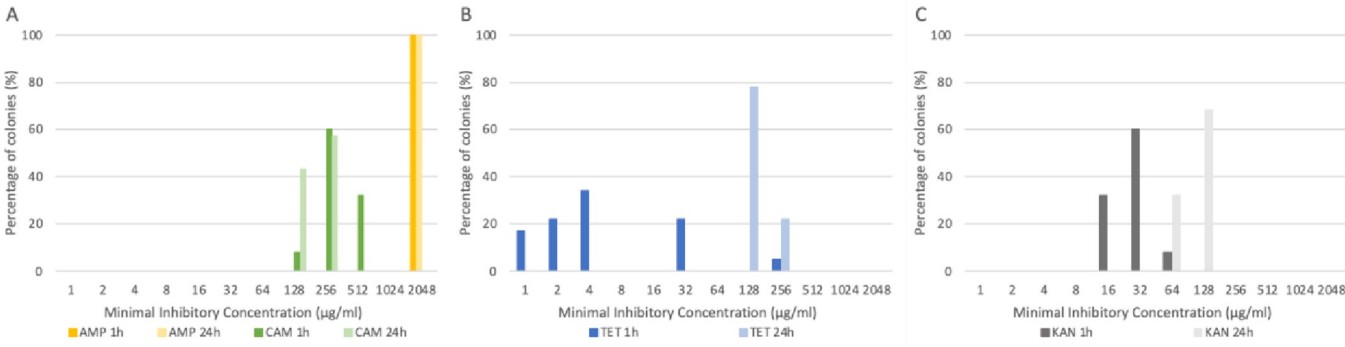

**Fig 4.** The percentage of isolated colonies found with a specific minimal inhibitory concentration (MIC) for antimicrobials: ampicillin and chloramphenicol (**A**), tetracycline (**B**) and kanamycin (**C**) after co-incubation of *E. coli* MG1655 and *E. coli* ESBL2082 for 1 hour or 24 hours and plating on double selective plates with ampicillin and chloramphenicol.

transconjugants never surpass more than 3% of the whole population after one hour of co-incubation. The role of energy stems from the metabolic costs of expression of the conjugative machinery. Continuous expression is adverse for survival of the host [18]. Thus, plasmids have adapted to keep conjugative transfer low and only some cells in the population get triggered to initiate transfer when they might be in the presence of a plasmid free host [28, 29]. Still, plasmid transfer usually seems to be initiated quickly, within 1 hour of co-incubation as observed in 14 out of 17 cases. However, some strains take longer before successful transfer was established. Possibly the rate of transfer depended on the ability to form pili, and indeed, the electron micrographs demonstrated presence of pili at the investigated time.

Transfer rates are affected by the number of plasmids a cell possesses, as all strains with two or more plasmids can transfer quickly, while two strains with one plasmid transfer slower. Increased conjugative transfer in the presence of other plasmids in the cell delays plasmid extinction [30]. So, multiple plasmids in one cell could increase plasmid transfer rates to survive. The two strains having only one plasmid contained an IncF plasmid. The IncF plasmids are on average larger in size than other plasmids found in this dataset, 130 kb compared to 90–120 kb for IncI plasmids and 30–60 kb for IncX plasmids [20]. IncF plasmids have an approximately 400 times lower conjugation rate than IncX plasmids, but a marginally higher transfer rate of 2.5 times than that of IncI plasmids [9]. Conjugation rates of IncF plasmids have been studied more often than those of IncX and IncI plasmids. More hours of co-incubation resulted in the transfer of additional or different plasmids because the cells are longer in contact with each other. The rate for taking up a second plasmid was in some cases increased when compared to the transfer rate of the first plasmid [31].

Plasmid transfer was confirmed to be species specific as attempts to obtain transconjugants with other gram-negative or gram-positive bacteria as recipients were unsuccessful. This is in line with other studies demonstrating that plasmid transfer happens more to kin and closely related strains [10, 12]. However, this only seems to hold true for liquid mating rather than filter mating [9].

Growth rates of donor strains differed strongly, even though they were all *E. coli* strains. The newly obtained plasmids do not produce a metabolic burden to *E. coli* MG1655, because the acquisition of one or more plasmids does not reduce the growth rate. This indicates that these transconjugants can compete well with the recipient strain but would still be outcompeted in most cases (13 out of 17) by the donor strain population. In 4 out of 17 cases, the transconjugant strain would become the dominant population. A similar pattern was observed in *E. coli* harboring ESBL-plasmids [32]. In that study most plasmids did not reduce the host

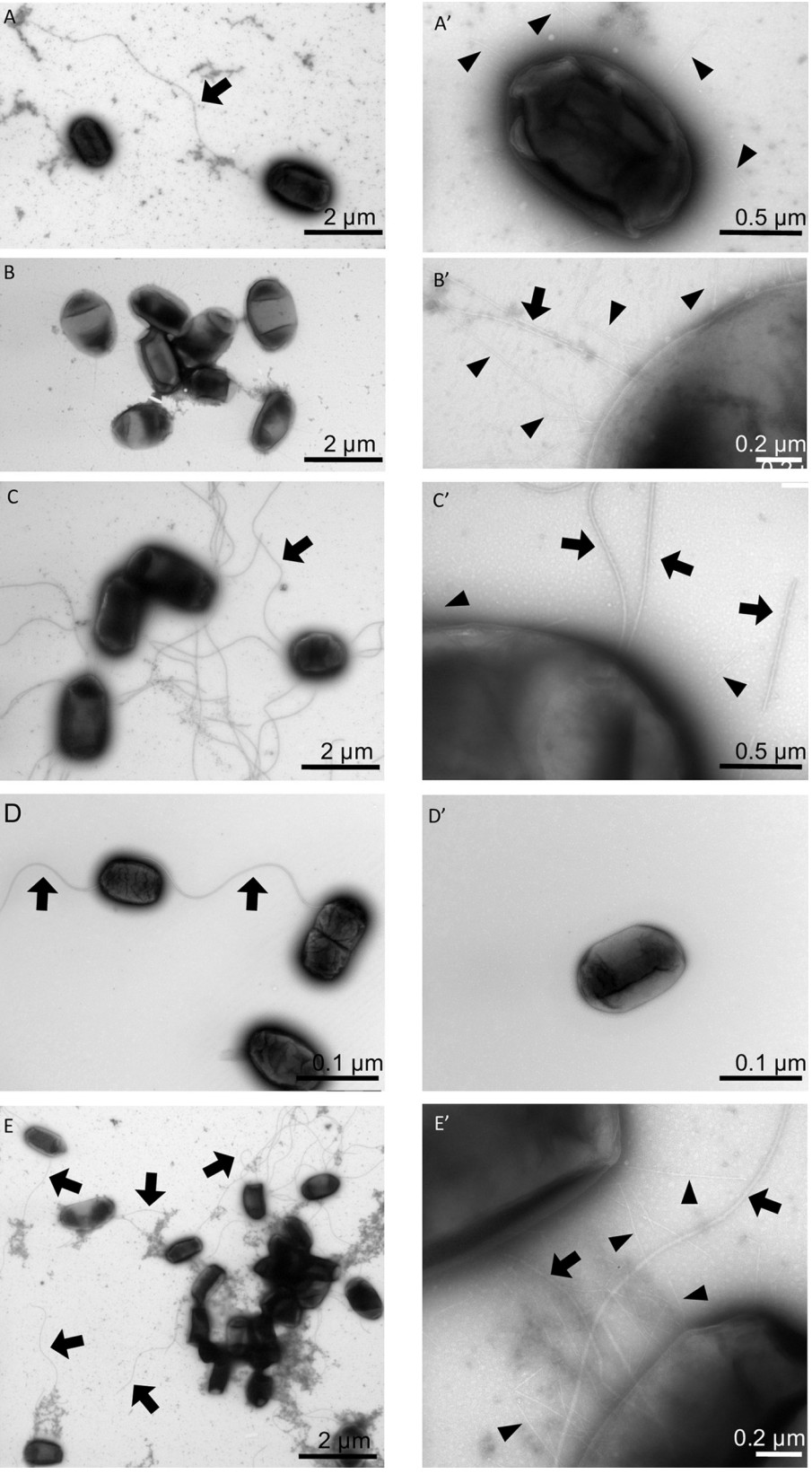

**Fig 5.** TEM micrographs taken from *E. coli* species ESBL2073 PFA fixed (**A**), ESBL3153 (**B**), MG1655 (**C**), Indole knockout (**D**) and E the mixture of ESBL3153 together with MG1655. In right column the close-up of pili (arrowheads) or flagellum (arrows) of same cells as in A-E.

growth rate significantly (8 out of 15). Only in some cases they grew less (4 out of 14) or more (2 out of 14). However, plasmids can burden the host just after the transfer event, before adapting over the first 24 hours towards a stable growth rate [33]. The plasmids thus effectively reduce their costs in a new host within the first 24 hours in order not to be outcompeted [28, 29, 32].

The comparison of the morphology of slow and fast mating cells using TEM suggests that aggregate formation and number of pili influence transfer rates. Pili are well known to have a role in the formation of aggregates. Aggregates and biofilms are hotspots for plasmid transfer [34–38], because increased cell-to-cell contact results in a higher chance of transfer events. The two donor strains compared in this study contained two similar plasmids, but one possessed an extra IncF plasmid. IncF plasmids can stimulate the formation of biofilms, which is after all a type of aggregate formation [39, 40]. Possibly the extra IncF plasmid in *E. coli* 3153 produces many pili, enabling a better formation of aggregates.

## Acknowledgments

We thank Kees Veldman and Ben Wit for providing the resistant strains and stimulating discussions. The students Mireille Nulkes, Siem de Haan, Veronica Zampieri, Ivan Fung, Eline de Waard, Sidra Kashif, Selin Alpagot, Eva Kozanli and Hannah Boersma performed transfer experiments, transformations, MIC, growth rate assays and TEM visualization as part of their degree or internship.

## Author Contributions

**Conceptualization:** Stanley Brul, Benno H. ter Kuile.

**Data curation:** Tania S. Darphorn, Belinda B. Koenders-van Sintanneland, Nicole N. van der Wel, Benno H. ter Kuile.

**Formal analysis:** Tania S. Darphorn, Nicole N. van der Wel, Benno H. ter Kuile.

**Funding acquisition:** Benno H. ter Kuile.

**Investigation:** Tania S. Darphorn, Belinda B. Koenders-van Sintanneland, Anita E. Grootemaat, Benno H. ter Kuile.

**Methodology:** Tania S. Darphorn, Belinda B. Koenders-van Sintanneland, Anita E. Grootemaat, Nicole N. van der Wel, Benno H. ter Kuile.

**Project administration:** Benno H. ter Kuile.

**Supervision:** Benno H. ter Kuile.

**Writing – original draft:** Tania S. Darphorn, Benno H. ter Kuile.

**Writing – review & editing:** Nicole N. van der Wel, Stanley Brul, Benno H. ter Kuile.

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
