## [Decision Letter · Decision Letter 0]

9 Feb 2022

PONE-D-21-39054Transfer dynamics of multi-resistance plasmids in Escherichia coli isolated from meatPLOS ONE

Dear Dr. ter Kuile,

Thank you for submitting your manuscript to PLOS ONE. After careful consideration, we feel that it has merit but does not fully meet PLOS ONE’s publication criteria as it currently stands. Therefore, we invite you to submit a revised version of the manuscript that addresses the points raised during the review process.

We look forward to receiving your revised manuscript.

Kind regards,

Hans-Uwe Dahms, Ph.D.

Academic Editor

PLOS ONE

Journal Requirements:

 [no]. 

Additional Editor Comments:

Pls. resubmit a REVISED VERSION of the MS considering all points raised by

2 reviewers as below in due time.

Reviewers' comments:

Reviewer's Responses to Questions

**Comments to the Author**

1. Is the manuscript technically sound, and do the data support the conclusions?

Reviewer #1: Yes

Reviewer #2: Yes

2. Has the statistical analysis been performed appropriately and rigorously? 

Reviewer #1: Yes

Reviewer #2: Yes

3. Have the authors made all data underlying the findings in their manuscript fully available?

Reviewer #1: No

Reviewer #2: No

4. Is the manuscript presented in an intelligible fashion and written in standard English?

Reviewer #1: Yes

Reviewer #2: No

5. Review Comments to the Author

Reviewer #1: PAPER REVIEW:

Resistance plasmids are critical for the spread of antibiotic resistance and are thus a topic of concern in veterinary and human medicine. Food borne Escherichia coli isolates containing one to five known plasmids were co-incubated with a generic recipient strain to examine plasmid transmission. Transfer rates were examined over extended co-incubation, across species, and during repeated back and forth transfer. Longer co-incubation resulted in the transmission of additional resistance types. The morphology of the interaction between co-incubated strains was studied using transmission electron microscopy. More pili interactions between cells result in improved aggregate formation and higher transfer rates.

Key Results given:

● Three donor strains were unable to produce transconjugants after 1 hour co-incubation, rather production was seen when incubated for 24 hours.

● Neither conventional nor LB aided glucose transformation procedures produced any transconjugates for the examined donor strains after 1 hour or 24-hour incubations.

● TEM pictures revealed numerous pili and flagella crossing the distance between two cells, supporting plasmid transfer.

● Higher resistance levels for tetracycline and kanamycin after co-incubation resulted in more plasmid transformation during prolonged co-incubation.

POSITIVE FEEDBACK:

● The test results obtained were effective and in coherent with the parameters that have been addressed with clear illustrations.

● Experiments were found to be conducted firmly.

● A standardized flow of data is examined.

CRITICAL COMMENTS:

● Time of plasmid transfer is not mentioned clearly; a specific time limit has to be mentioned for better clarity of the experiment.

● At what time interval range, analysis was performed for plasmid transformation, between 1 & 24 hours.

● Are these experimental data relevant or significant with higher resistance levels of other antibiotics similar to tetracycline and kanamycin.

● Data outcomes should be mentioned clearly in the discussion part in coherence with methodology used.

● Results should be concise and directly relevant to methodology used.

● The applicability and scope of this experimental data should be discussed in detail with ongoing studies concerning veterinary and human health .

Reviewer #2: Suggestions are provided in an appended version of the submitted MS. This includes some ENGLISH usage suggestions.

However, additional reading by an English expert is recommended.

Highest SIMILARITY (TURNITIN) was found with a copy sent to the UoAmsterdam for PLAGIARISM check. We

assume that this was submitted by one of the authors and recommend to DELETE this from the PLAG-platform!

Major flaws include:

● Time of plasmid transfer is not mentioned clearly; a specific time limit has to be mentioned for better clarity of the experiment.

● At what time interval range, analysis was performed for plasmid transformation, between 1 & 24 hours.

● Are these experimental data relevant or significant with higher resistance levels of other antibiotics similar to tetracycline and kanamycin.

● Data outcomes should be mentioned clearly in the discussion part in coherence with methodology used.

● Results should be concise and directly relevant to methodology used.

Other available data should be added as SUPPLEMENTS.

● A paragraph CONCLUSION should be added where potential applications of the study results are detailed with respect to improved:

- Experimental procedures

- Method innovation

- Governance of FOOD and DRUG policies

- others.

END

6. PLOS authors have the option to publish the peer review history of their article (what does this mean?). If published, this will include your full peer review and any attached files.

Reviewer #1: No

Reviewer #2: No

---

## [Author Response · Author response to Decision Letter 0]

21 Feb 2022

Line 63: “enables” refers to “machinery” not to “plasmids” 

Line 145: accepted and corrected accordingly. NB: after this correction all line numbers shift by 1. 

Line 229 (230) and line 257 (258): “suggests” is preferable over “suggested” because it is a conclusion that is not limited in time. 

Line 274: The flagella were visible in the EM pictures, but obviously not involved in plasmid transfer. We eliminated the mention of flagella as this was only confusing the issue. 

Line 296 (297): In our opinion this suggestion is not an improvement.

All other suggestions are incorporated in the new text.

---

## [Decision Letter · Decision Letter 1]

7 Jun 2022

Transfer dynamics of multi-resistance plasmids in Escherichia coli isolated from meat

PONE-D-21-39054R1

Dear Dr. ter Kuile,

We’re pleased to inform you that your manuscript has been judged scientifically suitable for publication and will be formally accepted for publication once it meets all outstanding technical requirements.

Kind regards,

Timothy J. Johnson

Academic Editor

PLOS ONE

Additional Editor Comments (optional):

Reviewer #1 brought up some additional suggestions for improvement. I will leave this to the author's discretion to make any additional changes in light of these comments. Because they were not brought up during the initial round of reviews, I do not think it is fair to require you to address them at this time. However, please look at them and consider them before your final publication.

Reviewers' comments:

Reviewer's Responses to Questions

**Comments to the Author**

1. If the authors have adequately addressed your comments raised in a previous round of review and you feel that this manuscript is now acceptable for publication, you may indicate that here to bypass the “Comments to the Author” section, enter your conflict of interest statement in the “Confidential to Editor” section, and submit your "Accept" recommendation.

Reviewer #1: All comments have been addressed

Reviewer #2: All comments have been addressed

2. Is the manuscript technically sound, and do the data support the conclusions?

Reviewer #1: Yes

Reviewer #2: Yes

3. Has the statistical analysis been performed appropriately and rigorously? 

Reviewer #1: I Don't Know

Reviewer #2: Yes

4. Have the authors made all data underlying the findings in their manuscript fully available?

Reviewer #1: Yes

Reviewer #2: Yes

5. Is the manuscript presented in an intelligible fashion and written in standard English?

Reviewer #1: No

Reviewer #2: Yes

6. Review Comments to the Author

Reviewer #1: Transfer dynamics of multi-resistance plasmids in Escherichia coli isolated from meat

Theidea of this research work is significant in the context ofeffectof the presence of multiple plasmids in E.coli in transferring resistance to recipient species via conjugation process.The overall test results done are effective and various parameters have been addressed with a clearrepresentation of the result.Appropriate concentrations of antibiotics used, OD values to be considered are clearly mentioned which can be effectively used as reference by other authors, universities and researchers. The provision of GenBank accession numbers of various strains adds more value to the text. Another important aspect that needs to be appreciated is the confirmation of the species specific nature of plasmid transfer.

In the methods section, there is more need to focus on themethod of isolation of resistant E.coli strains from meat.Why only chicken, turkey and bovine was used for plasmid isolation and why not mice (for example) or any other species? Another point that needs to be addressed is about the co-incubationtime as to why it has been conducted at only 1 hour and another for 24 hours. Why haven’t the authors tried to conduct for other time periods or gradient time interval range, or if they have conducted, then it should be mentioned in the methods section.

The size of the font set for figures 1 and 4 can be enlarged to an extent when compared with figures 3 and 4.

In line 52, ‘Resistance plasmids play a crucial rule’should be changed as ‘Resistance plasmids play a crucial role’

In line 102, ‘ResFinder to determine and confirm and find’ could be changed to ‘to determine, confirm and find’.

In line 179, ‘Before plating the transformation on selective plates’ is suggested for change as ‘before plating the transformed cellson selective plates’

Check on the overall grammar and flow of the content.

Otherwise, the article is overall well written and explained and can be advised for acceptance with minor corrections.

Reviewer #2: This contribution has now reached a level of detail, flow of thought, language improvement, and literature updated that

it becomes acceptable for our journal.

7. PLOS authors have the option to publish the peer review history of their article (what does this mean?). If published, this will include your full peer review and any attached files.

Reviewer #1: No

Reviewer #2: No

---

## [Editor Report · Acceptance letter]

23 Jun 2022

PONE-D-21-39054R1 

Transfer dynamics of multi-resistance plasmids in *Escherichia coli* isolated from meat 

Dear Dr. ter Kuile:

I'm pleased to inform you that your manuscript has been deemed suitable for publication in PLOS ONE. Congratulations! Your manuscript is now with our production department. 

Kind regards, 

on behalf of

Dr. Timothy J. Johnson 

Academic Editor

PLOS ONE